# Investigation on Maintenance Technology of Large-Scale Public Venues Based on BIM Technology

Tingchen Fang [1,2,3,*], Yiming Zhao [1,2,3], Jian Gong [1,2,3], Feiliang Wang [4,*] and Jian Yang [4,5]

1 Shanghai Construction Group Co., Ltd., Shanghai 200080, China; zhaoym@scgtc.com.cn (Y.Z.); gongjian_scg@126.com (J.G.)
2 College of Civil Engineering, Tongji University, Shanghai 200092, China
3 Shanghai Engineering Research Center of Super High Rise Building Intelligent Construction, Shanghai 200080, China
4 Shanghai Key Laboratory for Digital Maintenance of Buildings and Infrastructure, School of Naval Architecture, Ocean and Civil Engineering, Shanghai Jiao Tong University, Shanghai 200240, China; j.yang.1@sjtu.edu.cn
5 School of Civil Engineering, University of Birmingham, Birmingham B15 2TT, UK
* Correspondence: fangtc@scgtc.com.cn (T.F.); wongfayeleung@163.com (F.W.)

**Abstract:** Recently, the digital operation and maintenance of large-scale public venues have received increasing attention. The traditional building automation system (BAS), which can only provide information in a non-visualized way, is incapable of meeting the complex requirements of modern operation and maintenance. Therefore, a 3D-based building information modeling (BIM) technology is needed to improve operation and maintenance efficiency. In the paper, a BAS-to-BIM combined strategy is introduced, and the BIM-based maintenance object framework for large-scale public venues is re-built. The conversion method and lightweight method for the BIM maintenance model are introduced and a new type of public protocol, which can be used to attain a unified protocol layer that serves the BIM model, is proposed. In addition, this article presents the application of technologies, such as virtual/mixed reality, to improve the convenience of operation and maintenance. Finally, a practical project of a snow-sports stadium is given as an example to elaborate on the benefit of the proposed method. It indicates that the functions, for example, information integration, visualization, and positioning, introduced by BIM technology can effectively improve the quality and efficiency of project operation and maintenance.

**Keywords:** large-scale public venues; operation and maintenance system; BIM technology; public protocol; mixed reality

## 1. Introduction

Large-scale public venues, which are usually used for exhibitions, competitions, and conferences, have the characteristics of complex structural configurations, multiple functions, large flows of people/vehicles, complicated facilities and equipment, and high security requirements [1]. In order to evaluate and monitor the health status of large-scale public venues during the operation and maintenance stage, some advanced technologies, such as a building automation system (BAS), the Internet of Things (IoT), machine learning, edge analytics, and digital twin, have been proposed. Among them, a BAS provides a communication network, authentication protocols, access control, and integration for the operation and maintenance stage [2]. A BAS consists of all manual and automated management methods, involves the entire process from data collection, data analysis, to active control and provides fundamental support for power distribution, information release, parking management, anti-theft alarm, lighting control, etc. Therefore, the BAS has progressively developed as an important field of control systems during the operation and maintenance stage [3]. However, a BAS works as the fundamental system for monitoring

and control, and it does not cover the actual needs of structural operation and maintenance. This is because a BAS can only be used for manageable objects, such as wind systems, power equipment, but is incapable when it comes to objects that cannot be managed such as structural components. Operators who use the BAS should have some structural background, as all the information is based on two-dimensional drawings. Therefore, a BAS is unable to meet the requirements of visualized building operation and maintenance.

The introduction of building information modeling (BIM) technology can effectively solve the aforementioned issues and expand the application of a BAS. The application of BIM in the past decade developed from data collection to information integration and knowledge management [4]. BIM-based technologies offer efficient data management and exchange throughout the entire lifecycle of the project including construction quality management [5], building asset management [6,7], built environment monitoring [8,9], energy assessment [10,11], and waste estimation [12]. During the operation and maintenance phase, BIM technology utilizes object-based three-dimensional information models that innovatively improve traditional and inefficient 2D approaches for the planning, design, construction, and maintenance of infrastructure [13]. A BIM model can provide building managers with real-time data and information about energy consumption and the indoor conditions of buildings, but it also allows for viewing of the historical sensor data table and creation of graphical historical sensor data [14]. BIM can facilitate high-level analysis and evaluations for buildings by employing techniques such as acoustic analysis, carbon emission, construction and demolition waste management, operational energy use, and water use [15].

Moreover, in the life cycle of the building, the BIM model can be regarded as a database to be continuously expanded and improved. An extended interface of the 3D-BIM model needs to be established so that it can be connected and combined with other dimensions, e.g., to form an nD model that covers the construction and management of a project. Existing studies associated the dimensions of time (4D) [16], cost (5D) [17], sustainability (6D) [18], and facility management (FM) with 3D-BIM, that is, from the 3D-BIM model to the 7D-BIM model [19]. The utilization of 7D-BIM technology can provide optimized asset and data management of the building from design to demolition. Chen et al. [20] proposed an integrated BIM–FM methodology framework based on BIM technology and existing FM systems to improve the efficiency of facility maintenance management.

Despite the nD capability of BIM enabling its potential practice during building life cycle phases, designers–contractors focused primarily on the application of BIM during design–construction management stages [21]. Several gaps within the BIM for the FM domain need to be bridged, e.g., the lack of a framework, limited applications of compliance checking methods, limited use of open standards for the data sources required for the operational phase, and the lack of a prototype of a common data environment [22]. To date, BIM is still used as standalone information systems by building stakeholders, an integration technique including IoT sensors, facility management, BAS and BIM is needed for the end-user to monitor building performance data efficiently [23]. The information exchange between BAS and BIM requires an open communication protocol named Building Automation and Control network (BACnet) and the open BIM standard Industry Foundation Class (IFC) [24].

Therefore, it is necessary to develop a BAS-to-BIM integration technology for improving the operation and maintenance system. On the one hand, the BAS should be expanded to all virtualizable 3D objects, such as walls, structural beams, and columns; on the other hand, the 2D-based monitoring and control method in the BAS needs to be transferred to dynamic monitoring information along with the operation and maintenance process. In this paper, the framework of maintenance objects based on BIM technology is introduced. A new 3D-based type of operation and maintenance technology was established based on the traditional BAS for large-scale public venues. The application of virtual/mixed reality technique is introduced for better use, retrieve, and control of the BIM model under different scenarios. A case study based on a large snow-sports stadium is proposed to

explain the critical functions and maintenance objects of the BIM-based operation and maintenance technology.

## 2. Framework of Maintenance Objects

Determination of the maintenance objects is the first step when carrying out the operation and maintenance of large-scale public venues. From the perspective of the BAS, maintenance objects should refer to objects that can be monitored or controlled; from the perspective of BIM technology, maintenance objects should refer to all objects that meet the characteristics of 3D visualization. Compared to the BAS, the system based on BIM technology can increase the type and quantity of maintenance objects; hence, a new framework needs to be established.

Based on the above statement, maintenance objects can be divided into 3D objects, monitorable objects, and controllable objects. The 3D objects cover all the features of public venues and provide all the original static information, e.g., buildings, structures, and equipment. Monitorable objects are objects that can achieve feature data through technical means, such as the IoT, and generally refer to equipment. Controllable objects refer to objects that can be automated or manually controlled according to systematic decisions. When selecting controllable objects, it is necessary to divide the objects into groups, each group corresponds to a controllable object, and the monitoring data fusion can provide support for control decision making.

### 2.1. 3D Maintenance Objects

The 3D maintenance objects of large-scale public buildings should be classified according to classification standards, such as buildings, structures, heating ventilation air conditioning (HVAC), water supply and drainage, and electrical system, as shown in Figure 1. The building information is circulated inthe plan design phase, preliminary design phase, deepening design phase, construction phase, completion delivery phase, and operation and maintenance phase. The attributes of the maintenance object increase with increasing phases and the attributes generated in the early phase may be entirely covered by the latter phases. Therefore, it needs to be considered whether the information at each phase needs to be retained, e.g., for use in the operation and maintenance phase.

For building objects, from the perspective of the BAS, the spatial information and usage information of all rooms are mainly required, whereas from the perspective of BIM technology, the overall dimensions and location information of the building site, road, and building are mainly required, as well as the shape features and location information of equipment, fixed furniture, and architectural decorations.

For structural objects, from the perspective of the BAS, dynamic information, such as the internal force and deformation of the main components is required; but from the perspective of BIM technology, the actual size and position of various types of structural components are mainly required including steel structures, walls, beams and columns, structural floors, roofs, and ramps.

For HVAC objects, from the perspective of the BAS, the quantities, parameters, materials, connection methods and maintenance and replacement records of HVAC equipment are required; but from the perspective of BIM technology, the geometric information and position of HVAC equipment with pipelines are mostly required.

For water supply and drainage objects, from the perspective of BAS, the engineering quantity, material, and maintenance and replacement records of the water supply and drainage system and fire protection system are mainly required; but from the perspective of BIM technology, it is mainly required to obtain the dimensions, spatial distribution, geometric information of the water supply and drainage system, and also the path planning information of the fire protection system.

For electrical objects, from the perspective of BAS, engineering quantities, system types, parameters, materials, connection methods, and maintenance and replacement records of the electrical equipment are required; but from the perspective of BIM technol-

ogy, it is mainly required to obtain the geometric information, positioning, and spatial distribution of the electrical equipment with pipelines.

For all the above objects, the traceability of the operation and maintenance information should be considered, including but not limited to the information in the construction or installation process as well as repair and replacement information.

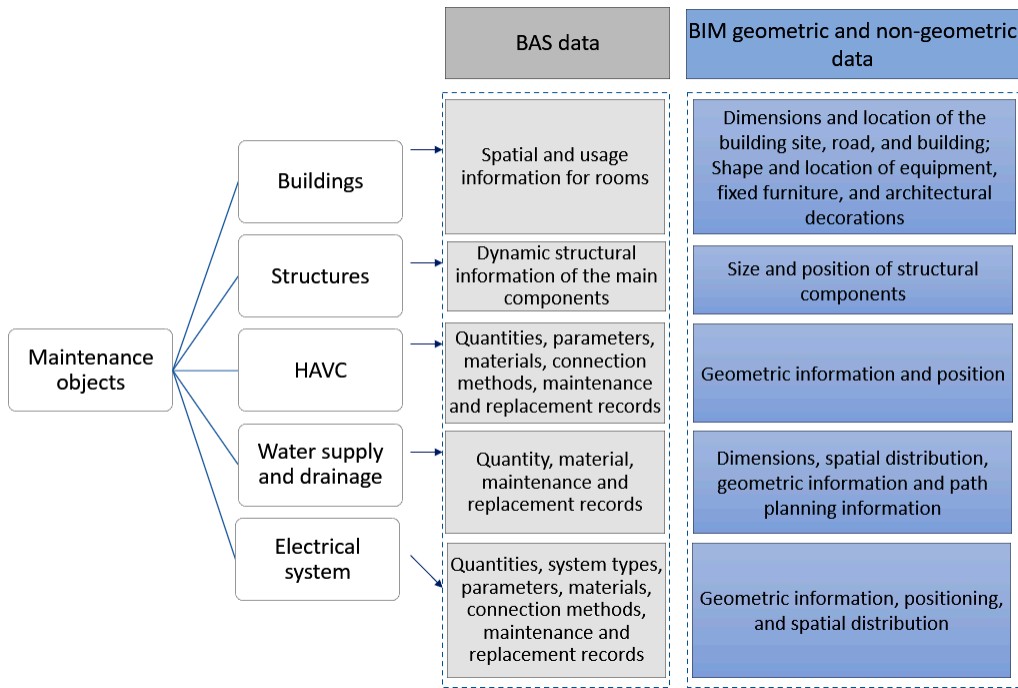

**Figure 1.** Composition of maintenance objects.

## 2.2. Monitorable and Controllable Objects

Both monitorable and controllable objects belong to the traditional BAS, which can collect data through various types of sensors and realize automatic control through control instructions. Limited by the items that can be collected by the sensors, the objects that can be monitored mainly include environmental data (e.g., indoor temperature, indoor humidity, air quality, wind speed and wind direction, rainfall, and water quality), video data (e.g., the flow of people and vehicles), and structural data (e.g., internal force and deformation of the building). It should be noted that structural data mainly uses sensors embedded in the structure during the construction period for data acquisition such as strain sensors for measuring structural stress and level sensors for measuring structural settlement.

Control instructions for controllable objects can be fulfilled based on analysis of the monitoring data. However, it is difficult to control a certain device using a single type of data. To obtain an effective control decision of a device, analysis needs to be conducted based on a large amount of various data. In the operation and maintenance of large-scale public venues, the temporary switch of HVAC is often comprehensively controlled through indoor temperature, indoor humidity, and air quality, and the water supply and drainage system is effectively controlled through data such as rainfall, water quality, and water consumption. Moreover, the load input and response data of the building can be obtained through the monitoring data of people and vehicles and based on the measured internal force and deformation of the structural components. With the up-to-date finite element model, a safety warning load can be given when people and traffic conditions exceed the limit value. The specific logic is shown in Figure 2.

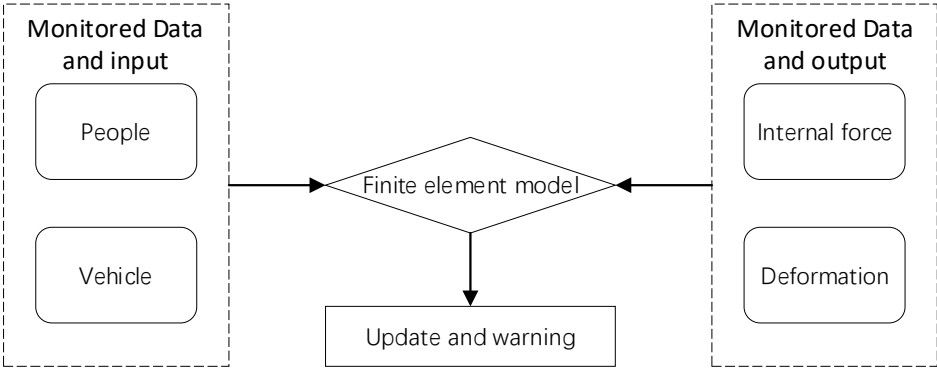

**Figure 2.** Specific logic for update and warning of the finite element model.

A complete data loop is formed by combined monitorable objects and controllable objects. After the control measures are completed, positive feedback of the monitoring data should be obtained to indicate that the characteristics represented by the monitoring data are developing in a better direction. Therefore, it is required that the monitoring data and control instructions have an accurate time sequence, that is, the monitoring data and control instructions are required to be reliable, stable, and timely, and the time delay is generally required to be controlled within 20 ms. For large-scale public venues, both wired transmission and wireless technology based on the IoT can be used to fulfil the high-speed transmission of monitoring data and control. The wired transmission method is relatively reliable, but the wiring cost is high, and the maintenance is complicated; while the wireless transmission equipment is simple to install and convenient to maintain. Therefore, the wireless transmission method has been widely promoted and applied in recent years.

## 3. Maintenance Technology Based on BIM

To apply BIM technology in the operation and maintenance system for large-scale public venues, new maintenance technology based on BIM was studied herein. This study included the establishment of the BIM maintenance model, the lightweight and online loading of the BIM maintenance model, the design of a data/command transmission protocol [25] related to the BIM model, and the application of virtual/mixed reality technology applications [26], etc.

### 3.1. Establishment of BIM Maintenance Model

When establishing a BIM maintenance model, the most important concepts that need to be clarified are the level of details (LODs) and model information depth. LOD refers to the level of details of the geometric and non-geometric information of the model elements, and model information depth refers to the authenticity and accuracy of the geometric and non-geometric information when the model elements are visually presented. Generally speaking, LOD determines the amount of information of the entire model, and model information depth determines the amount of information that the entire model can display, that is, LOD plays a decisive role in the model information depth. But for dynamical information, the model information depth will continue to increase and update, which will further optimize LOD. Therefore, for a well-functioning BIM maintenance model, its LOD will continue to improve, and the model information depth that can be displayed will also continue to expand.

The maintenance model is converted from the as-built delivery model, which can be obtained by combining the BIM, construction model, and Asset Information Model (AIM) [27] as shown in Figure 3. The completed delivery model should reach the accuracy level of LOD400 or LOD500 and should include various professional sub-models, viewports, data tables, and data documents, auxiliary audio, and video files that can be bound to the model.

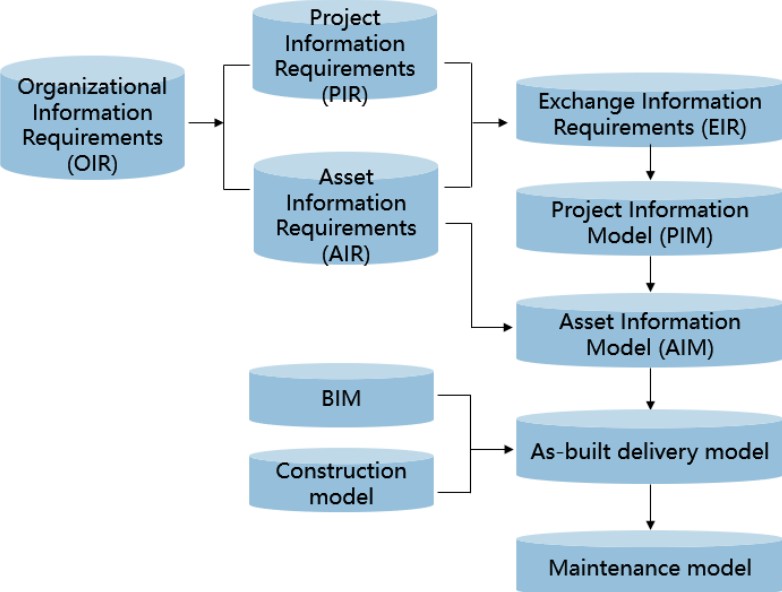

**Figure 3.** Relationship between the model in different phases.

The difference between the maintenance model and the as-built delivery model is not only in model accuracy but also in the composition of the basic units in the model. The as-built delivery model is the integration of various professional sub-models under a unified coordinate system, focus on all the components of the model, and can only serve the construction phase. But the maintenance model is used to assist the operation and maintenance of the building and pay more attention to the logical relationship between different sub-models, also between sub-models and external environments. After the application of the maintenance model, a large amount of dynamic data will be integrated with time such as the race schedule, event records. Therefore, the maintenance model can perform a better and more direct service in the process of operation and maintenance when compared to the as-built delivery model. Moreover, it is necessary to present a method to achieve conversion from an as-built delivery model to a maintenance model, and it is better to be an automated method with consideration of conversion efficiency.

The maintenance model can be converted through the following steps. Firstly, reorganize the components in the as-built delivery model, to form three types of sub-models including single-floor and full-professional sub-model, single-floor and single-professional sub-model, and single-professional sub-model. Secondly, conduct the separation of figure model and information; the information can be serialized and recorded in the database. Thirdly, clarify the necessary maintenance objects, and delete or disable the unnecessary components in the database. Finally, perform the logical relationship processing between different single-professional sub-models on the same floor and between the same single-professional sub-model on different floors. To realize automatic conversion, the above steps can be specifically described as:

(1) Read the information on floors and professional sub-models. Split the as-built delivery model based on the height of building floors. The principle of splitting is to split the floor first and then the professional sub-models, and finally build into the above three types of sub-models. Reading information and splitting process can be automatically performed through the API interface of the model file;

(2) Automatic separation of figure model and information. Taking the RVT format file in Autodesk Revit as an example, the component and data in the model can be divided using the Autodesk Forge Platform, and the uniform id of each component can be acquired. Based on the uniform id and other characteristic information, a globally unique identifier (GUID) can be defined as the unique identifier of the component in the global context. GUID and corresponding information can be stored in a relational

database such as MySQL. The component can be stored in a common surface model format file such as FBX format. Take the room as an example, record the separated room data in the database including the contour lines, component types, component positions, component sizes. All data are serialized into strings and recorded in the database, and the room's 3D model is saved as a figure model.

(3)　A new operation and maintenance system is established to carry maintenance models. The operation and maintenance system includes single-floor and full-professional subsystem, single-floor and single-professional subsystem, and single-professional subsystem. Each subsystem corresponds to a type of model. Establish the mapping relationship among the above three subsystems and the components in the operation and maintenance model; delete and disable all components without a mapping relationship.

(4)　Establish the discontinuity points of the adjacent cross-floor professional subsystems. The discontinuity information can be obtained through the automatic comparison of the spatial distribution of the three types of subsystems and should be stored in the database. The points must also be established with a GUID number. For example, the electromechanical logic between different floors should be serialized into a character string and then recorded in the database, including the position of the discontinuity point, the connection direction, and the connection path.

(5)　Establish the logical relationship between cross-professional subsystems. Since the professional subsystem of large-scale public venue can be divided into categories, such as water, electricity, wind, and fire protection, the logical relationship of cross-professional subsystems can be realized by pre-customization. The automated constructed logical relationship will also be stored in the database, and each logical relationship is stored as a complete record. For example, the logical relationship between the building's power supply equipment and the electrical equipment, the relationship between the air-conditioning water system and the air-conditioning wind system, etc. This logical relationship can be among multiple subsystems on a single floor or among multiple subsystems on multiple floors.

In addition, after the maintenance model is established, it is necessary to check the logical relationship generated to ensure that it is accurate; and it is necessary to add points on the model according to the BAS, to form a three-dimensional coordinate point or monitoring area to connect dynamic monitoring data. These two parts of work can be completed manually in the model.

### 3.2. Lightweight and Online Loading of BIM Maintenance Model

The lightweight operation of the maintenance figure model can be completed by two measures: firstly, the BIM solid model can be converted into a general surface model format such as FBX, and the surface model generation parameters are controlled during the conversion process dynamically, to reduce the number of triangular facets in the model; secondly, to store the same building, structure, and decorative components, etc., in the processed figure model in the same manner, which can reduce the redundant number of the same components, thereby greatly reducing the overall number of components in the model. All the lightweight processes of the model can be carried out automatically without manual intervention.

After the aforementioned lightweight operations, the model size of conventional public venues will be reduced by 70–90%. The first method mentioned above can reduce the size by approximately 50%. The second operation can further reduce the size by approximately 30%. It can provide strong support for fast and efficient online loading of models during operation and maintenance. The online loading of the model can be done through WebGL technology including the definition of the scene, camera, renderer, and loading of the entire model into the scene.

### 3.3. Design of Data Transmission Protocol Related to BIM

Traditionally, the data transmission protocol is mainly used for the BAS, and it provides standard support for the acquisition of monitoring data and the sending of instructions. However, the operation and maintenance system based on BIM technology requires higher standards on the protocol. It must fully consider the characteristics of the BIM model and form a monitoring and control mechanism with the combination of the BAS and BIM models. In the data transmission process, the BAS is regarded as the bottom layer and the BIM model should be used as the core expression. All the monitoring data must be displayed in a 3D visualization based on the model, and all the control instructions can be realized through the model. Therefore, the information of the model should be considered in the design of the protocol.

To ensure that monitoring data are transmitted to the system in a timely manner and that control instructions can be transmitted to terminal equipment at high speed, it is necessary to ensure real-time and reliable data transmission. Between the device terminal and the BAS, commonly used protocols include BACnet, OPC, Modbus, RS485, etc. In addition, some terminal devices also provide API interfaces for the BAS. However, it should be noted that these protocols only consider the transmission of data, and it is impossible to meet the abovementioned requirements for adding model information to the protocol. For this reason, based on the abovementioned protocol, a new public standard protocol for conversion is proposed in this research.

The basic format of the public standard protocol is shown in Figure 4. It has 10 sections, with a total length of 30 + N bytes. The explanation of each section is as follows.

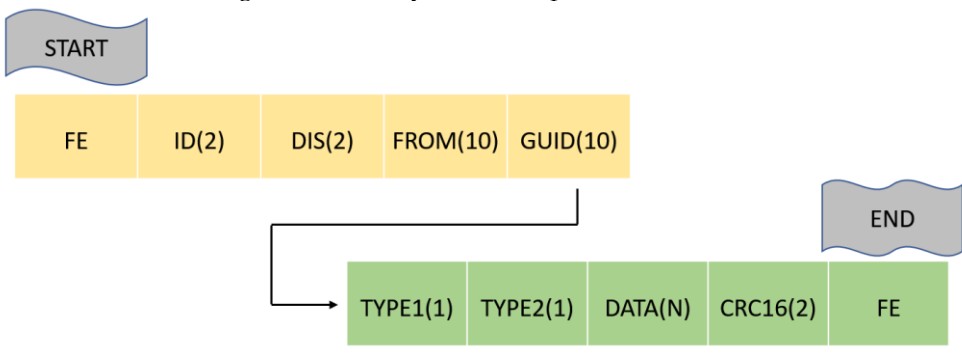

**Figure 4.** The basic format of the public standard protocol.

1.  The first "FE" section means the start point of the single data package, while the last "FE" section means the endpoint;
2.  The "DIS" section has 2 bytes, which represents the location of terminal equipment, and it is mainly used for mobility monitoring equipment, such as a mobile vehicle for environmental monitoring, and if the equipment is stationary, then write "00 00";
3.  The "FROM" section has 10 bytes, which stands for the sender's number, and it is used to clarify the path of the instruction, and if the data package is not an instruction, then write as "00";
4.  The "GUID" section has 10 bytes and can be used to clarify the BIM component. It is noted that, for the consideration of the size of the data package, the size of GUID here is smaller than the GUID stored in the database; the component can be uniquely determined only with 10 bytes in the GUID;
5.  The "TYPE1" section only has 1 byte, and it stands for the type of the data package. When the data package is a control instruction, write "01"; and when the data package is monitoring data, write "02"; and when else, write "00" instead;
6.  The "TYPE2" section also has 1 byte, and it stands for the type of data. For example, if the data is information of deformation, write "01"; if the data is room temperature, write "02". This section can be divided according to the actual situation. In the

snow-sports stadium case involved in this paper, there were 35 data types divided, which covered all existing data types;

7.  The "DATA" section has N bytes, and it stands for dynamic data, including the monitoring data and control instruction data. The specific value of N is determined with the "FROM", "TYPE1", and "TYPE2" sections, and different parameter combinations have a different N number. The analysis of "DATA" also refers to the same principle, finding and applying the analysis rules according to the above three parameters;

8.  The "CRC16" section has 2 bytes, and it stands for the check of the data package. Through this parameter, the receiver can judge the correctness and integrity of the data packet.

The above-mentioned protocol defines a set of unified standards for docking with the system, and its application methods include two types: (1) the user can modify the original protocol of the hardware terminal to the public protocol and send it directly to the system or (2) based on all third-party protocols, write a software interface for protocol conversion and convert it to a public protocol for sending. The case involved in this paper adopts the second scheme. The main reason is that the first scheme requires protocol update operations on the hardware. The workload is large and there are many uncertain influencing factors. For example, once the GUID information in the database is changed, the information recorded in the hardware will also be changed accordingly.

Among all data packets, the packets related to control instructions have the highest real-time requirements. To increase the computing power and improve real-time performance, the edge computing method of the local computer was adopted. The method was applied using multiple industrial control computer equipment or single-chip microcomputer equipment in the entire local area network environment. The information transmission of these devices is based on the abovementioned protocol. Moreover, to maintain the stability of the entire network, a fault monitoring and early warning mechanism for network channels between all devices should be established. The main method is that each terminal device separately monitors the unblocked conditions of the upstream and downstream networks, and the monitoring results can be summarized to the system using the detection instructions for network stability analysis, and the detection instructions also following the protocol mentioned above.

### 3.4. Virtual/Mixed Reality Technology Applications

Virtual reality technology, also known as VR technology, is used to provide a fully virtualized 3D scene so that 3D models can be loaded in the virtualized scene, and the operator's perspective is also placed in it [28]. In general, VR technology often uses static 3D scenes and 3D models to simulate various accidents, equipment operation procedures, etc., [29]. By using VR technology, operators can quickly and efficiently complete training by directly operating the virtual equipment, and do the operation and maintenance work in advance, and realize the seamless transition of as-built, take-over, and maintenance stages. VR technology can be associated with the real physical world through software interfaces in the actual operation and maintenance of public venues. It can realize the direct remote operation of the equipment in the real physical world by the operator in the VR scene. The technical architecture is shown in Figure 5, and the realization is shown in Figure 6.

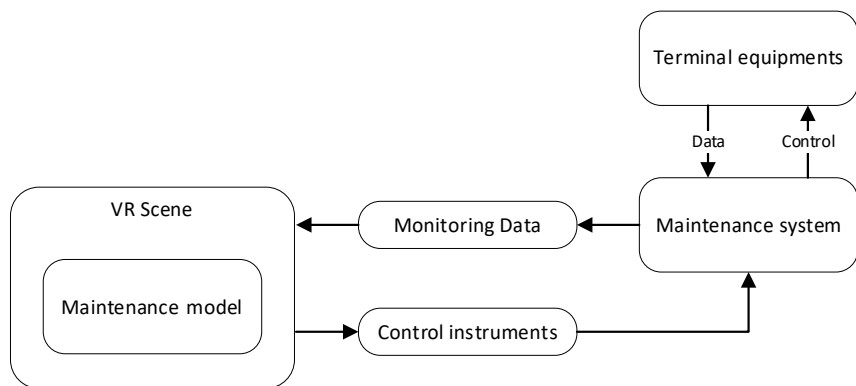

**Figure 5.** The architecture of VR technology.

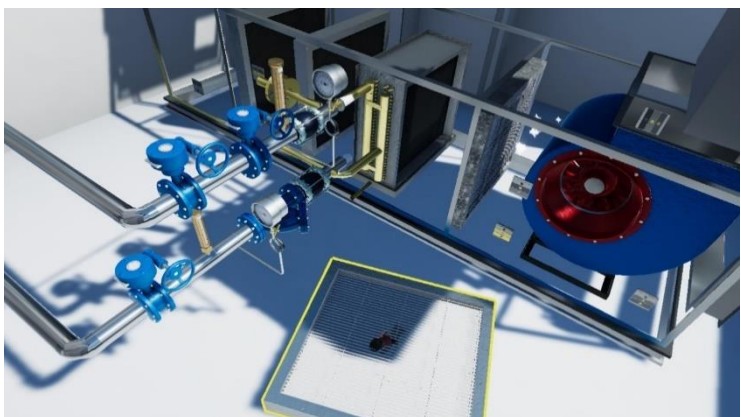

**Figure 6.** Realization of VR technology.

However, VR technology has a critical imperfection, that is, it needs to provide a 3D virtual model which must be completely loaded into the VR scene. This process could be costly in terms of time and money and leads to a limitation for the use of VR technology. Therefore, a new mixed reality (MR) technology is proposed to solve this problem.

The MR technology can project 3D objects into the real world, and the model is dynamically changeable. In the operation and maintenance process of large-scale public venues, two methods can be used for MR technology. One is the hooking method, which directly hooks objects to the real world such as pipelines to the real-world building or component. The other is the combination method that combines different objects into a new 3D scene placed on a real-world plane and performs monitoring and control operations inside the combination. In this way, the method can be conducted without being restricted by the static model in VR technology. The technical architecture is shown in Figure 7 and the realization is shown in Figure 8. As shown in the architecture, with the MR technology, objects can be directly manipulated in the real world and feedback to the MR scene in real-time if the objects can be monitored. This is the biggest difference between MR and VR technology.

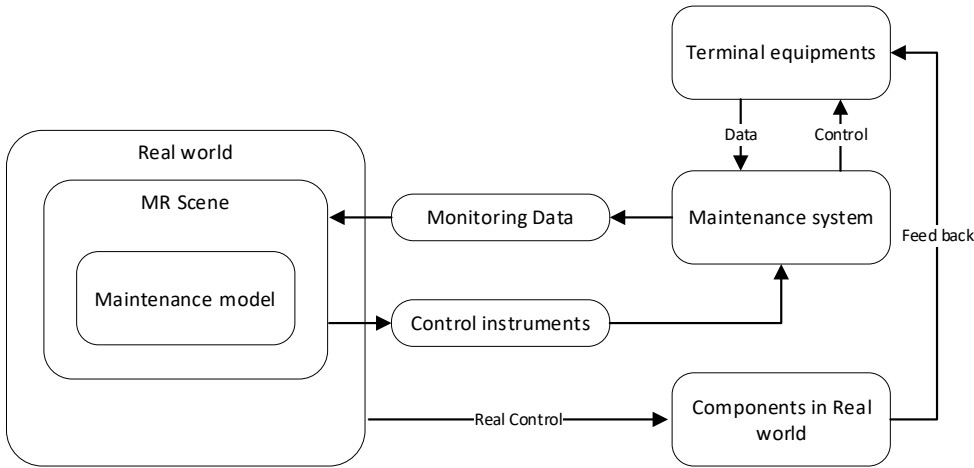

**Figure 7.** The architecture of MR technology.

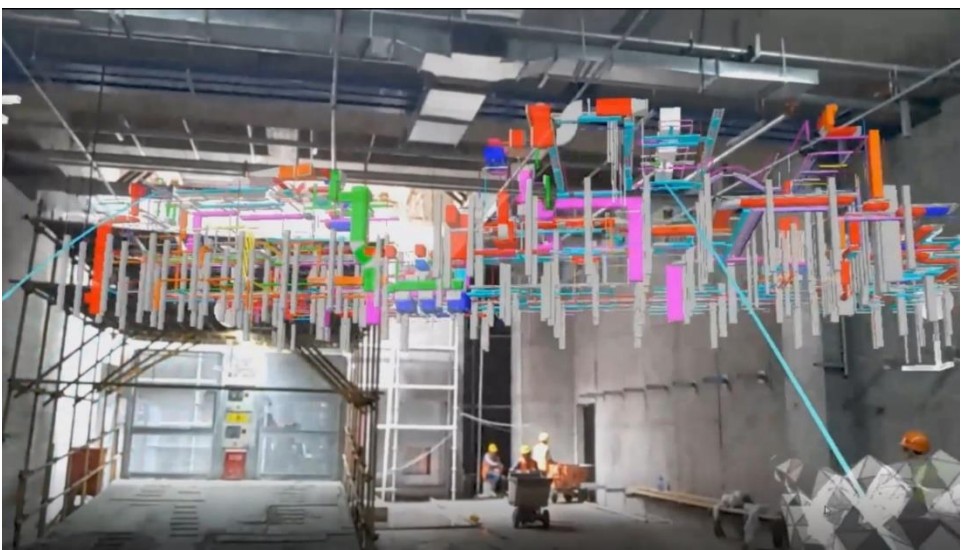

**Figure 8.** Realization of MR technology.

It should be noted that for both VR and MR technologies, the key to application is to synchronize the data in the virtual world and the real world. That is, all operations in the VR scene or MR scene should be transmitted to the terminal equipment in the actual world through the protocol, and the data collected by the terminal equipment should also be fed back to the virtual scene in real-time. Compared with VR technology, MR technology provides a much more profound control method, which can recognize finger movements and operate through eye movement, while VR technology is mainly completed by positioning in space, handle the driving, button clicking, and scene switching. Therefore, MR technology has relatively better application prospects.

In the use of VR/MR technology, it is necessary to record the application process, and the video recorder is the best logging method. By collecting picture frames of VR/MR scenes in real-time and composing a video stream in the form of picture frames, it can be used as a basis for future traceability. For this kind of video stream, a file storage format is developed which can be used to store a large amount of video frame data and can quickly read any one of the picture frames as shown in Figure 9. This file has three levels of the index, including time, object, type, and can index all picture frames. After testing, using this file format as storage, a single file can reach a maximum of 100 GB, and the retrieval time of a single picture frame can be within 1s, which greatly improves the storage and retrieval efficiency.

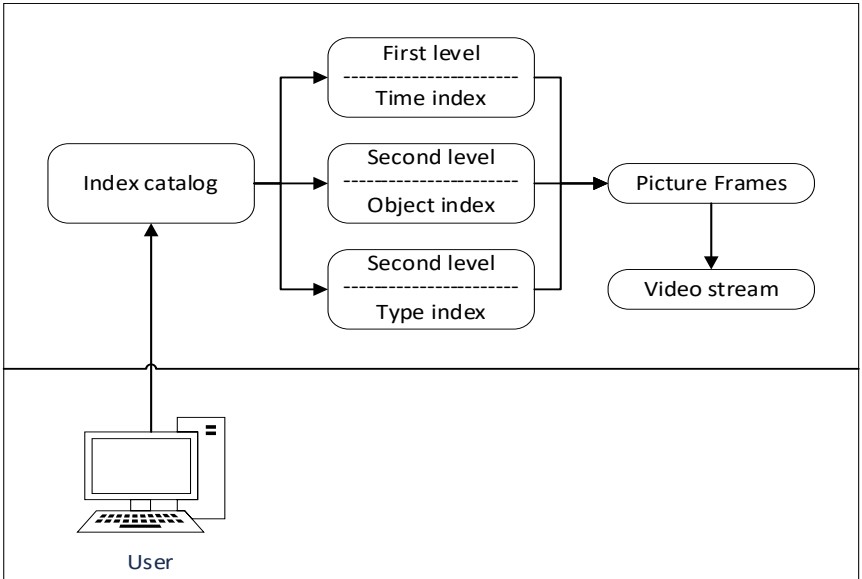

**Figure 9.** A file storage format used for the recording of the VR/MR scene.

## 4. Operation and Maintenance System for Snow-Sports Stadium

Based on the above studies, the operation and maintenance system for a snow-sports stadium is presented herein as an example. The system consists of a four-layer framework as shown in Figure 10. The first layer is the data layer, which refers to the terminal devices and data collection. The second layer is the protocol layer, which converts the data packet from third-party devices into a standard public protocol and sends it to the third layer. Meanwhile, the second layer receives various control commands created by the third layer and sends them back to the devices. The third layer is the traditional BAS layer for data analysis and decision-making during operation and maintenance. The fourth layer is the 3D BIM visualization layer, which carries the BIM maintenance model, and can operate the BIM model and information in a visualized way.

BIM technology can be well compatible with the FM system to achieve a comprehensive operation and maintenance of the building. After the completion of the project construction stage, the as-built BIM model can provide a reference for the data applied in the operation and maintenance stage, which lead to a life-cycle information circulation. The function of the BIM model can be embodied in information integration, visualization, and rapid positioning of building components (see Figure 10): (1) For the operation and maintenance information integration, the building information generated in the design and construction stage involves many participants, and the information in the traditional transmission process is easy to lose. Through the BIM transmission protocol, the building information generated in the design and construction stage can be completely transferred to the operation and maintenance stage, avoiding the loss or repeated entry of information. (2) In terms of visualization, previous AutoCAD drawings, which are used by property managers and customers, can be replaced by a 3D model based on BIM technology. Therefore, the whole project model and critical parameters can be browsed more conveniently, and a great reduction in the management's difficulty can be predicted for the users. (3) During the operation and maintenance, inspection or replacement of structural components and equipment should be carried out regularly. Meanwhile, the confirmation of the location of the target components or equipment is crucial, and it will directly affect the management's effectiveness and customer experience. Components and equipment can be easily located using the BIM model, and maintenance personnel can quickly query the basic information of the object, such as the manufacturer, maintenance records, to facilitate rapid decision making during the maintenance process. After the maintenance is completed, the maintenance record will be uploaded to the BIM model to provide the basis for the subsequent maintenance work.

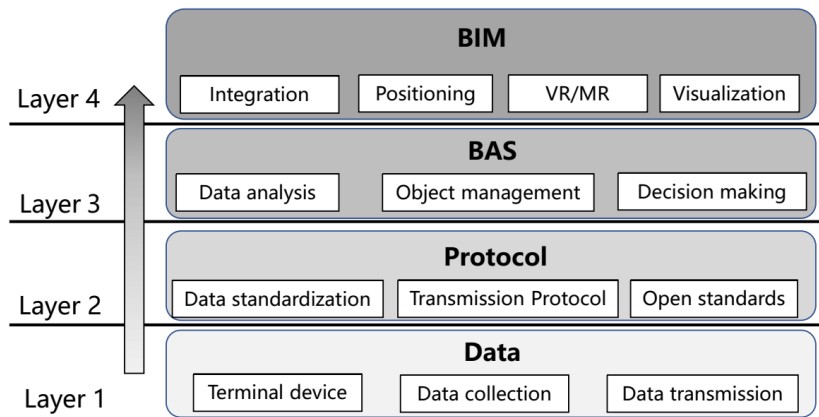

**Figure 10.** The framework of the proposed operation and maintenance system.

The BIM platform also contains detailed information for structural safety monitoring, energy consumption control, and space and equipment management as shown in Figure 11. For structural safety monitoring, IoT terminal devices are mainly used to monitor key parameters of the structure, such as deformation, and vibration and declination, and edge analysis and model inversion are used to evaluate the safety of the overall structure. In terms of energy consumption control, the BIM model is used to generate statistics on the energy consumption of the whole building in different periods, including electricity, and water and gas consumption. Based on data analysis, energy-saving strategies are formulated for managers to reduce the cost of operation and maintenance. In terms of equipment management, facilities, and equipment, such as water pumps, fresh air pipelines, and hydrant facilities, can be checked at any time through the BIM operation and maintenance platform, and related equipment attributes and maintenance records can be directly displayed. Users can clearly understand the operation of each air conditioner, fan, pipe, and valve as shown in Figure 12a. When the alarm message appears, the equipment position can be highlighted, and the operating parameters, manufacturers, manual, warranty period, etc., are listed in the model simultaneously. For space management, different colors are used to mark spaces for different purposes in the BIM model. Through label definition, spatial positioning and attributes can be found and edited in the model, to provide efficient services for long-term and variable tenancy management as shown in Figure 12b.

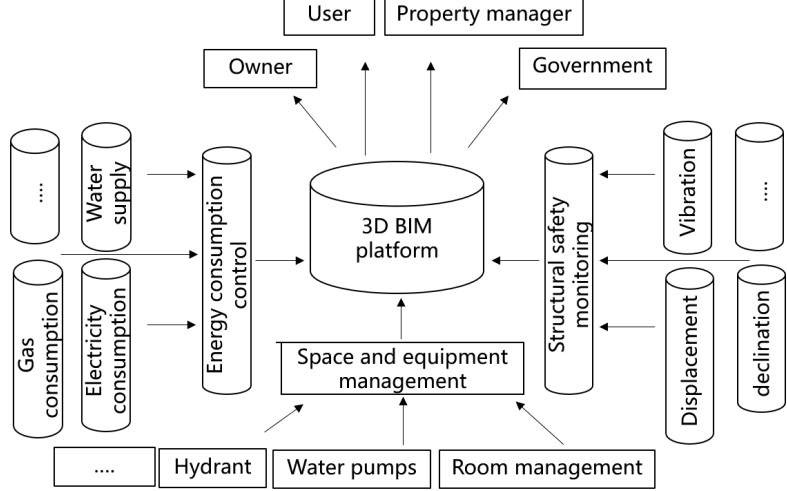

**Figure 11.** Integration in the maintenance system.

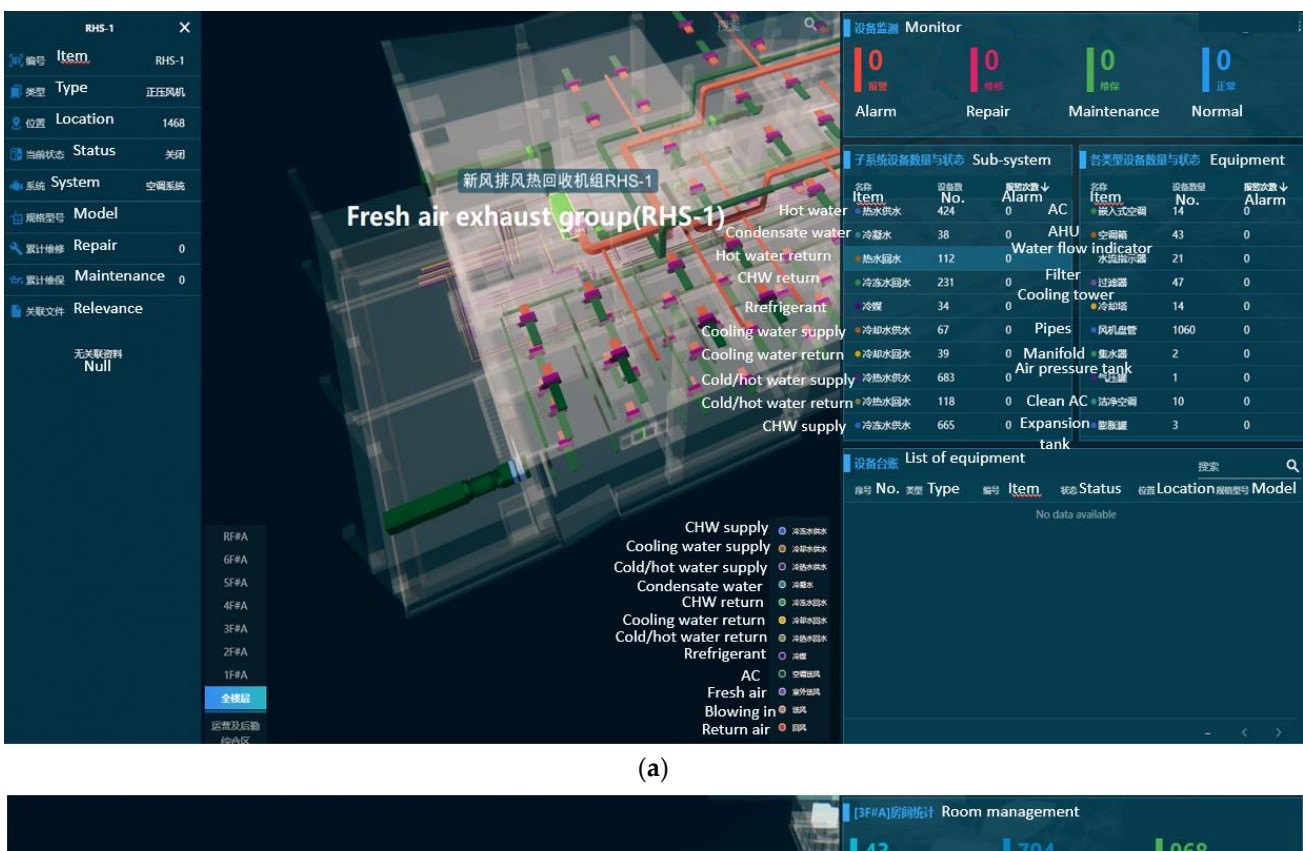

(**a**)

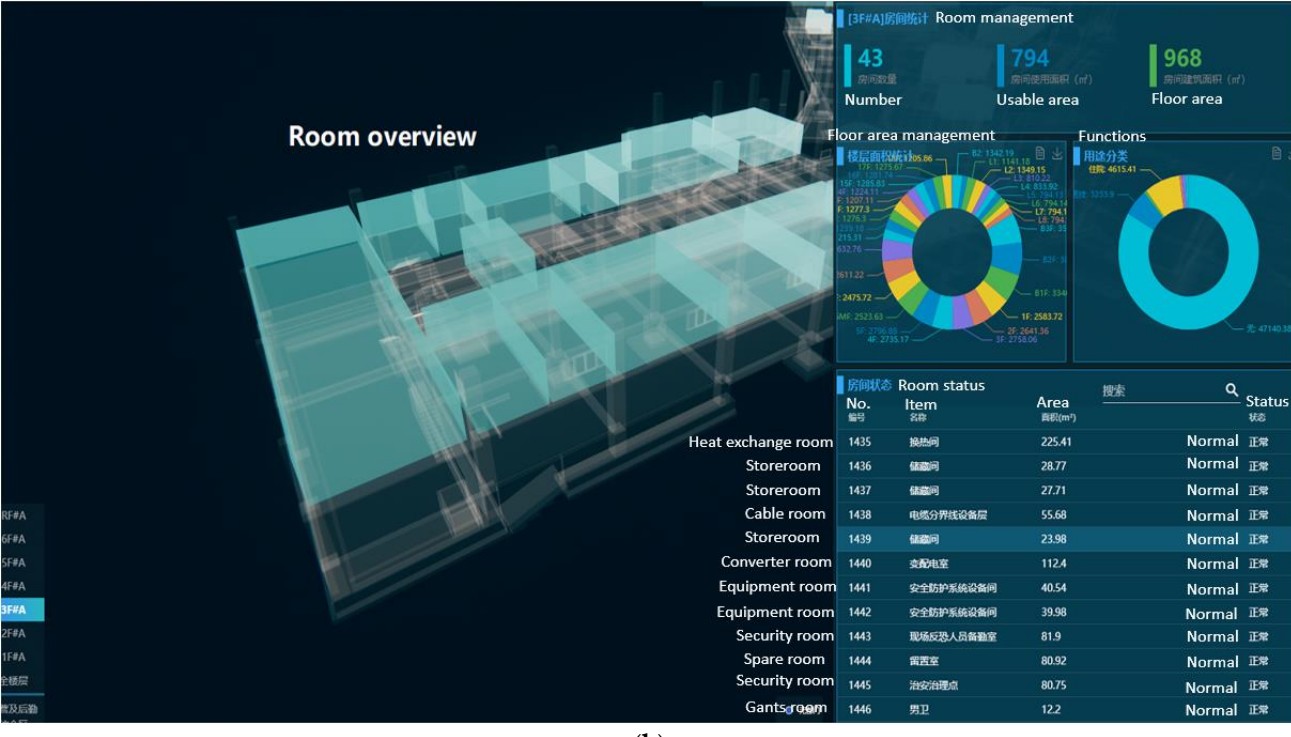

(**b**)

**Figure 12.** Operation and maintenance system for snow-sports stadium: (**a**) functional modules for fresh air systems based on BIM; (**b**) functional modules for room management based on BIM.

Some special maintenance objects need to be noted for the snow-sports stadium project, e.g., geographic information, sports facility objects, and auxiliary facility objects.

1.   Geographical information objects. Divide the geographical area with the racetrack as the object, obtain the topography, landform, scope, and other characteristics of the

geographical area; obtain the location of various facilities in the geographical area within and outside the race schedule;

2. Sports facility objects. Obtain information, such as the shape, size, direction, and position of the track as well as the wind speed, ambient temperature and humidity, illuminance, night lighting. In addition, the maintenance information of the track at different times should be obtained in a timely manner. For small-scale competition facilities, like a referee tower and a display screen of match results, only basic shape and location information is needed;

3. Auxiliary facilities object. Facilities, such as a snowmaking system, need to be focused and maintained frequently. The production type, manufacturer, operation method, materials, position, information of regular maintenance and regular use should be obtained.

It should be noted that for the special maintenance objects of snow-sports stadiums, different maintenance strategies depend on the stage. During a race, it is necessary to obtain the dynamic information of the objects and conduct maintenance in real-time, while after a race, only regular maintenance is acquired.

The as-built delivery model is shown in Figure 13, and a converted maintenance model is shown in Figure 14.

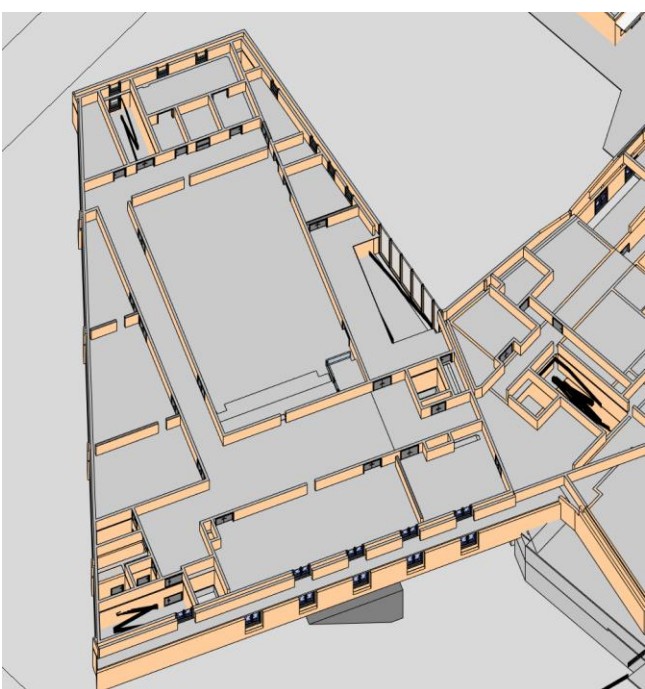

**Figure 13.** As-built delivery model.

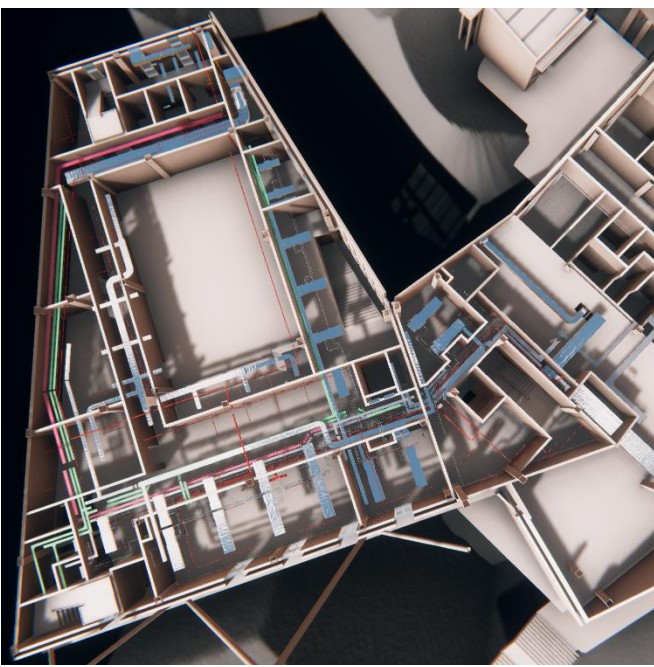

**Figure 14.** Converted maintenance model.

The protocol layer of the snow-sports stadium mainly targets building component objects, equipment objects, and monitoring equipment objects. The monitoring equipment objects include wind speed and direction equipment, environmental temperature and humidity equipment, video monitoring equipment, water quality monitoring equipment, power consumption monitoring equipment, voltage stability monitoring equipment, indoor temperature and humidity monitoring equipment, air conditioning controllers, lighting systems, parking systems, snowmaking equipment control systems, multimedia control systems, fire control systems, door opening and closing control systems, elevator control systems. As mentioned earlier, all types of third-party protocols are uniformly converted into the same protocol through the conversion of the protocol layer and sent to the operation and maintenance system for use; the instructions issued by the system are also converted to different manners according to the protocol layer and drives the controllable object to change.

The system also designed four types of access permissions, mainly for owners of the stadium, users such as tourists and athletes, property maintenance staff, and government departments concerned with the stadium. All the functions of the system are accessible to the property maintenance staff, while part of the data are open to the other three types of authority.

## 5. Conclusions

In this study, a BIM-based operation and maintenance system for large-scale public venues was investigated. A new maintenance object framework and a new data transmission protocol based on the combined BAS and BIM model was developed. Moreover, the conversion procedure, lightweight method, and VR/MR technology for the BIM maintenance model in the operation and maintenance process were discussed. The following conclusions can be drawn from the study.

- The conversion of the maintenance model including re-assembling the components in the as-built delivery model, separation of the figure model and information, deletion or disabling the unnecessary components in the database, and performing the logical relationship processing between different sub-models;
- The lightweight operation of the maintenance figure model can be completed by two steps, i.e., to convert the BIM solid model into a surface model format, such as FBX,

and to reduce the redundant number of the same components by storing the building, structure, and decorative components with the same configuration in the same way. After the lightweight operations, the model size of conventional public venues will be reduced by 70–90%.

- A new data transmission protocol for the BIM-related operation and maintenance system is proposed. It is suggested to write a software interface for protocol conversion and convert it to a public protocol for sending, as the method is efficient without the requirements of hardware protocol updating.

**Author Contributions:** Conceptualization, T.F. and J.G.; writing—original draft preparation, T.F. and Y.Z.; writing—review and editing, F.W. and J.Y. All authors have read and agreed to the published version of the manuscript.

**Funding:** The research was supported by the National Key R&D Program for the 13th Five-Year Plan of China (2018YFF0300301 in 2018YFF0300300).

**Institutional Review Board Statement:** Not Applicable.

**Informed Consent Statement:** Not Applicable.

**Data Availability Statement:** Not Applicable.

**Conflicts of Interest:** The authors declare no conflict of interest.

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
