# Peer review of "Investigation on Maintenance Technology of Large-Scale Public Venues Based on BIM Technology"

_sustainability, doi:10.3390/su13147937_

Round 1

Reviewer 1 Report

The topic discussed in the article is actual and very necessary, both from the scientific and practical side. However, the article is in my opinion underdeveloped and requires a few changes and additions.

1. Firstly, in the Introduction, the authors only mention individual publications, without discussing them. Listing articles for the purpose of "artificially" increasing the quantity of quotes should be avoided. This section should be discussed in more detail by presenting specific citations relevant to the article, not just claims that technologies such as BA, IoT, etc. are used. This can be written without quoting. I suggest making a part of an introduction and citing (by giving specific sentences / quotes, not just listing them) those publications that are able to contribute to the article.

2. A few editing errors, e.g. Point 3 is written in a different font; tab. 1 is shifted to the left in relation to the text, fig. 9 is incorrectly signed (a and b should be in the description of fig. 9, etc.

3. Fig. 3 - If the authors write about BIM, then in Fig. They should use terms related to BIM. Instead of "design model" it should be "BIM model", as biuld model is the LOD 500 used in the text. In addition, the term maintaince model depends not only on "as build model", but on many investor requirements. The hierarchy of organizational requirements is given e.g. in ISO 19650 standards (I recommend reading) - so you can define the organizational requirements of OIR and asset information requirements AIR affecting the Asset Information Model (AIM) and the impact of PIM (Project Information Model). I suggest supplementing the drawing with informative content and names used in the world.

4. BIM 7D is used by managers in process of the object management on the stage of its operation. 7D allows participants of the process to extract and store data assets, such as .: the state of the object / component, technical specifications, required maintenance schedule and technical reviews, manuals or applicable warranty period. Such an approach to the facility management process not only improves the whole process, but also improves the quality of services in this area. This concept is not mentioned in the entire article - 7D, which is closely related to the maintenance phase, and this is what the article is about.

5. The authors themselves write about the need to take into account geometric and non-geometric data, and they focus only on the presentation in the form of ornaments (drawings) of geometric information, the idea of BIM is not based on 3D presentations, but on the circulation, collection and processing of information. What information should be included in the 7D model? What should be updated - the authors write nothing about it. The part related to the information content of the model definitely needs to be supplemented.

6. Similarly, case study sets. No information contained in the model is shown, there is no information about the flow of data and, importantly, about their use - what the authors of BIM introduce ...
This part seems to be added for the sole purpose of showing visually appealing pictures ...

The article requires solid changes and rethinking the concept of its presentation. At the moment it is not suitable for printing, but I see the possibility of its significant improvement.

Author Response

The authors would like to thank the reviewers for their comprehensive and constructive comments. These comments have been carefully considered and revisions have been made to the manuscript. Responses to these comments and revisions implemented in the paper are detailed below.

The topic discussed in the article is actual and very necessary, both from the scientific and practical side. However, the article is in my opinion underdeveloped and requires a few changes and additions.

  1. Firstly, in the Introduction, the authors only mention individual publications, without discussing them. Listing articles for the purpose of "artificially" increasing the quantity of quotes should be avoided. This section should be discussed in more detail by presenting specific citations relevant to the article, not just claims that technologies such as BA, IoT, etc. are used. This can be written without quoting. I suggest making a part of an introduction and citing (by giving specific sentences / quotes, not just listing them) those publications that are able to contribute to the article.

Thanks for the valuable suggestion, the introduction section has been revised according to the comment (Page 1 Line 36-46, Page 2 Line 64-92).

  1. A few editing errors, e.g. Point 3 is written in a different font; tab. 1 is shifted to the left in relation to the text, fig. 9 is incorrectly signed (a and b should be in the description of fig. 9, etc.

Thanks. The mentioned errors and typos have been addressed.

  1. Fig. 3 - If the authors write about BIM, then in Fig. They should use terms related to BIM. Instead of "design model" it should be "BIM model", as build model is the LOD 500 used in the text. In addition, the term maintaince model depends not only on "as build model", but on many investor requirements. The hierarchy of organizational requirements is given e.g. in ISO 19650 standards (I recommend reading) - so you can define the organizational requirements of OIR and asset information requirements AIR affecting the Asset Information Model (AIM) and the impact of PIM (Project Information Model). I suggest supplementing the drawing with informative content and names used in the world.

Thanks for the comment. Figure 3 has been redrawn according to ISO 19650 (Page 6 Line 227).

4. BIM 7D is used by managers in process of the object management on the stage of its operation. 7D allows participants of the process to extract and store data assets, such as: the state of the object / component, technical specifications, required maintenance schedule and technical reviews, manuals or applicable warranty period. Such an approach to the facility management process not only improves the whole process, but also improves the quality of services in this area. This concept is not mentioned in the entire article - 7D, which is closely related to the maintenance phase, and this is what the article is about.

Thanks, more explanations have been added in the manuscript (Page 2 Line 70-80).

  1. The authors themselves write about the need to take into account geometric and non-geometric data, and they focus only on the presentation in the form of ornaments (drawings) of geometric information, the idea of BIM is not based on 3D presentations, but on the circulation, collection and processing of information. What information should be included in the 7D model? What should be updated - the authors write nothing about it. The part related to the information content of the model definitely needs to be supplemented.

Thanks, more explanations have been added in the manuscript (Page 12-13 Line 460-483).

  1. Similarly, case study sets. No information contained in the model is shown, there is no information about the flow of data and, importantly, about their use - what the authors of BIM introduce ...
    This part seems to be added for the sole purpose of showing visually appealing pictures ...

Thanks for this suggestion, more information has been added in the manuscript (Page 13 Line 487-506).

The article requires solid changes and rethinking the concept of its presentation. At the moment it is not suitable for printing, but I see the possibility of its significant improvement.

Reviewer 2 Report

Dear authors,

After reading your paper, I don’t se framework that you proposed. Please make a diagram or flow chart of your framework. Introduction isn’t written well. Are there some protocols developed? If are what is advantage of the one you propose? Also, you propose new standard protocol for data transmission so I don’t see how is that connected with sustainability?

Best regards, reviewer

Author Response

The authors would like to thank the reviewers for their comprehensive and constructive comments. These comments have been carefully considered and revisions have been made to the manuscript. Responses to these comments and revisions implemented in the paper are detailed below.

  • After reading your paper, I don’t see framework that you proposed.
    Please make a diagram or flow chart of your framework.

Thanks for the suggestion, a flow chart of the framework has been added in the manuscript (Page 13 Line 485 Figure 10).

  • Introduction isn’t written well. Are there some protocols developed? If are what is advantage of the one you propose? Also, you propose new standard protocol for data transmission so I don’t see how is that connected with sustainability?

Thanks, the introduction has been rewritten according to the comment (Page 1 Line 36-46, Page 2 Line 64-92).

Reviewer 3 Report

  1. As the construction industry have been adopting the BIM methodology, a study concerning an approach to support its implementation in maintenance, based in the developed of an innovative BA-to-BIM strategy, is presented. The BA subsystem is well know but in BIM context in novelty. Useful and relevant study.
  2. The problem to be resolve in clearly present in the first tem as well as the structure of the text. The key of the strategy is based in “maintenance objects can be divided into 3D objects, monitorable objects, and controllable objects” retrieved from a BIM model generated according to the objectives required in the maintenance activity, in a wide range of public buildings.
  3. In “Figure 1. Composition of maintenance objects” the areas are to large. Did the authors subdivided the domains building, structures, . What elements of buildings o structures were considered?
  4. In “Figure 3. Relationship between the model in different phases” the design model and the construction, requires some attention to the way of modelling both, as they must present distinct segmentation or explosion of components in of its composite materials. These problems were not analyzed properly?

Author Response

The authors would like to thank the reviewers for their comprehensive and constructive comments. These comments have been carefully considered andbrevisions have been made to the manuscript. Responses to these comments and revisions implemented in the paper are detailed below.

  • As the construction industry have been adopting the BIM methodology, a study concerning an approach to support its implementation in maintenance, based in the developed of an innovative BA-to-BIM strategy, is presented. The BA subsystem is well know but in BIM context in novelty. Useful and relevant study.

Thanks.

  • The problem to be resolve in clearly present in the first tem as well as the structure of the text. The key of the strategy is based in “maintenance objects can be divided into 3D objects, monitorable objects, and controllable objects” retrieved from a BIM model generated according to the objectives required in the maintenance activity, in a wide range of public buildings.

Thanks.

  • In “Figure 1. Composition of maintenance objects” the areas are too large. Did the authors subdivided the domains building, structures. What elements of buildings or structures were considered?

Thanks for this valuable comment, Figure 1 has been redrawn according to the suggestion (Page 4 Line 161).

  • In “Figure 3. Relationship between the model in different phases” the design model and the construction, requires some attention to the way of modelling both, as they must present distinct segmentation or explosion of components in of its composite materials. These problems were not analyzed properly?

Thanks. Figure 3 has been redrawn according to the comment (Page 6 Line 227).

Round 2

Reviewer 1 Report

The authors corrected the article according to my suggestions. The background of the article in introduction could possibly be improved, but in my opinion the article is ready for print.

Reviewer 2 Report

Dear Authors,

Thank you for making all changes according to my recommendations.

I have no further questions.

Best regards, reviewer

Reviewer 3 Report

The comments were adequately attended